# Mechanistic Interpretability analysis of a single-layer transformer on 0-1 knapsack

## Abstract

Small lanugage models have been shown to exhibit generalisation for toy problems while being trained on algorithmically generated datasets. It is poorly understood whether this phenomenon happens in complex problems such as NP-complete problems. In this work, we show the inability of a single-layer transformer to "grok" the 0-1 knapsack problem. We analyze the internals using visualisations and interpretability techniques and show why the model is not able to form a robust internal circuit. This shows how transformer-based models struggle to generalize on NP-complete problems as well as their inability to solve problems requiring high amount of computation. This work showcases why LLM-based AI agents should not be deployed in high-impact spaces where a vast amount of planning and computation is required.

## 1 Introduction

The last decade has seen rapid progress on many difficult problems using machine learning (ML) and artificial intelligence (AI). This has included high-impact scenarios such as autonomous vehicles as well as decision-making for bail in criminial courts (Hamilton & Ugwudike, 2023). Recently, ChatGPT and other popular large language models (LLMs) have become some of the most widely used forms of AI (Mitra et al., 2023). Traditionally, complex systems are tested and deployed only after guarantees are achieved on their safety - such as aircrafts, and even the atomic bomb (Wiescher & Langanke, 2024). However, as seen in the existing literature, the community still doesn't have a complete understanding of LLMs (Anwar et al., 2024). As a result, it is irresponsible and dangerous to continue the development and deployment of LLMs without sufficient work on understanding on its internals.

Mechanistic interpretability is a sub-field of ML interpretability that dissects the behaviour of individual components of a transformer and their interconnections (Räuker et al., 2023). It goes beyond explainable-AI (XAI) in not just providing correlations, but also uncovers actual causal mechanisms (Olah et al., 2020). Mechanistic interpretability also helps in providing trustwhory ways to understand the model's internals (Hsia et al., 2023). Existing studies in literature have only focused on toy problems ((Nanda et al., 2023), (Zhong et al., 2024), (Quirke & Barez, 2023), and (Chughtai et al., 2023)). They tend to focus on P problems, and do not explore other complex algorithmic tasks. Therefore, we focus on understanding the ability of single-layer transformers to "grok" NP-complete problems.

In this study, we analyze the attention patterns of the trained small-scale transformer, and hypothesize why it is unable to perform well. We conclude by stating the implications of this work for the mechanistic interpretability landscape.

## 2 Experiments and Observations

We use a single-layer transformer setup using the TransformerLens library (Nanda & Bloom, 2022). We train these transformers to solve the 0-1 knapsack problem, where the transformer has to give the best possible price as output.

Although we intially considered an dataset which had relatively high variance as seen in Figures 1 and 2 (Chauhan, 2022), we switched to an algorithmically generated dataset based on work showing

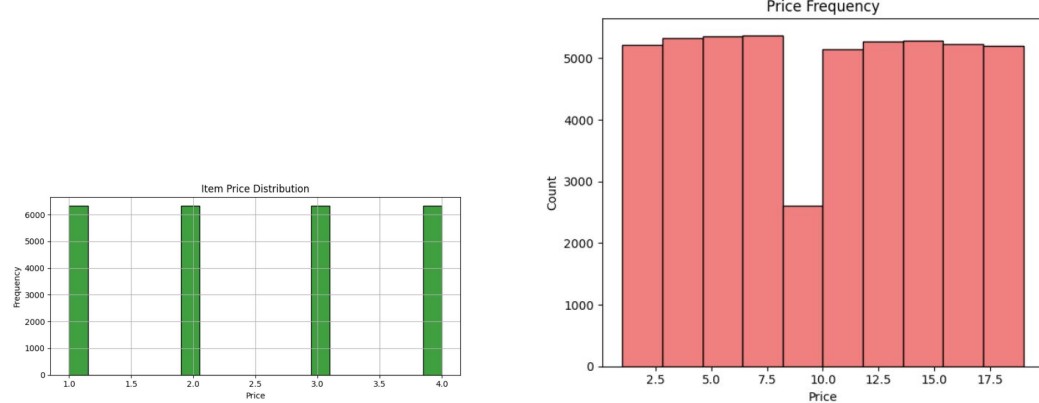

Figure 1: Frequency distribution of the prices of the items in the two datasets

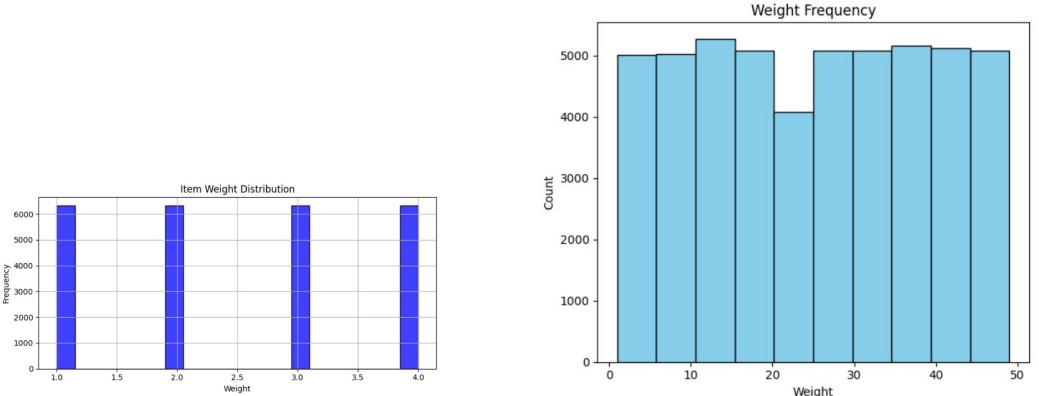

Figure 2: Frequency distribution of the weights of the items in the datasets

how networks can reliably exhibit grokking on small systemically generated datsets (Power et al., 2022). We constrain our dataset to only contain 4 objects due to compute constraints. The dataset is configured as follows: $W_1, W_2, W_3, W_4, P_1, P_2, P_3, P_4, C$ and $BP$, where $W_i$ represents the weight of the $i^{th}$ object, $P_i$ represents the price of the $i^{th}$ object and $C$ represents the capacity of the knapsack. $BP$ represents the best possible worth of items whichcan be placed in the knapsack of given capacity $C$. We set the weights and prices to be all permutations of the range $1, \ldots, n$. The capacity of the knapsack contains all possible unique sums possible from the superset of $\{1, \ldots, n\}$.

We train the model using the AdamW optimizer, and continue the training upto 100k epochs. However, the model was unable to grok. We studied why the model was unable using a combination of visualisations and mechanistic interpretability techniques.

We can see from the attention weights visualisations that the model places more importance on the capacity token than any other token. It also places relatively more importance on the price tokens than the weight tokens. We also use singular values analysis to compare the embedding matrix of the trained model with that of a matrix with the same shape. We find that the both matrices have relatively similar graphs, showing that the amount of variance captured by each principal component is not better than a matrix with random values. We also compare this to the singular values of the embedding matrix of a model trained on modular subtraction, which shows a sharp drop off in variance after the first few components.

Similarly, we look at the variance of the principal components, which shows how the matrix doesn't have any smooth sinusoidal patterns. We hypothesize that this could be due to two factors, either the the knapsack problem having a more complex and less transparent embedding space, or the inability of our model to capture the task's underlying structure.

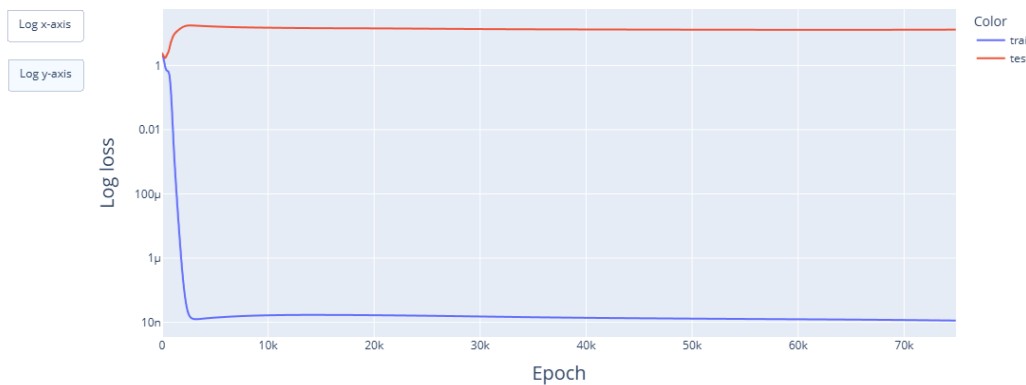

Figure 3: Train and test log-loss curve

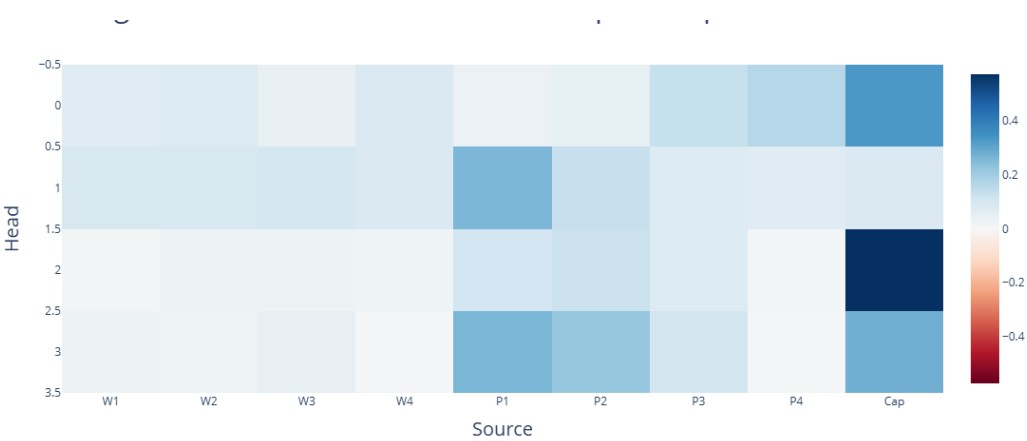

Figure 4: Average attention patterns across all input samples

Logit lens is used to investigate how the model's predictions evovle as information progagtes through its layers (nostalgebraist, 2021). Even though we only use a single-layer model in this study, here we examine the model's output at different processing stages, such as after the embedding layer, after the attention layer, and after the multi-layer perceptron (MLP) layer. By projecting these intermediate representatiosn back into the output space, we are able to understand the relative importance of each of the model's components. We find that the MLP layer has the highest impact in shaping's the model's decision.

Probing is a mechanistic interpreatibility technique which is used to understand the ability of the model to store accurate representations of the data (Ju et al., 2024). We train a linear regressor to predict the given input based on the internal representations, and we find that the model is able to perfectly store upto half of the weights and prices. However, it struggles to accurately form representations of the other weights and prices, as well as the capacity of the knapsack.

We also use activation patching (Heimersheim & Nanda, 2024) to find that activations of the neurons attendning to the capacity token, have a relatively high impact on the loss. This shows that the model is highly dependent on the capacity constraint to model its output.

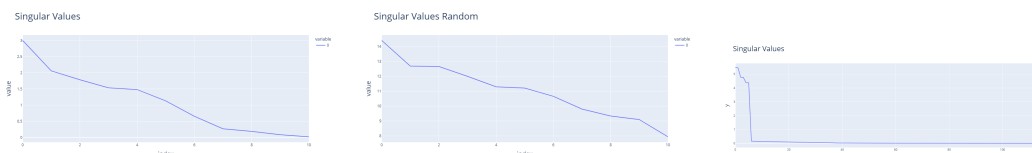

Figure 5: Singular values comparison of our model, a random matrix, and a model trained on modular subtraction

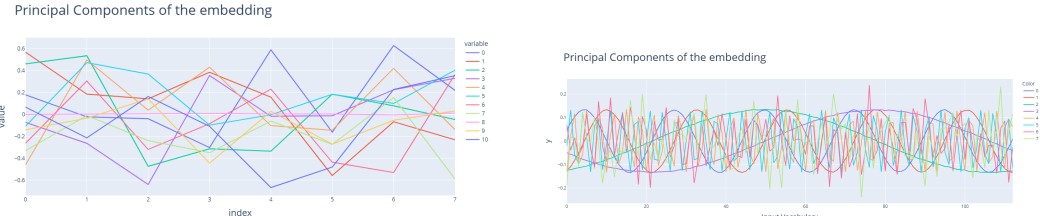

Figure 6: Variation of principal components of our model, and a model trained on modular subtraction

## 3 CONCLUSION

We are able to observe a lack of robustness in the model's ability to model an algorithm for solving the knapsack problem. In particular for this problem, we find that the model struggles to generalize due to its failure to effectively integrate the capacity constraint during the initial embedding phase, as the well as the lack of layers, which restrict the power of the model.

We hypothesize that:

1. Transformer-based models struggle to generalize to NP-complete tasks due to the **combinatorial explosion** in considerations involved.
2. Transformer-based models with $k$ layers will only be able to generalize to tasks which can be solved using $O(n^k)$ time complexity algorithms.

This raises major doubts about the ability of LLM-based AI systems to reliably act as agents, since we find a fundamental lack of ability to generalize. We also find how the models can still give *believable* answers, even though they do not have the proper internal circuits to process the data. Therefore, further work is needed to limit the exposure of LLM-based AI systems to tasks which involve planning and computation through regulations and laws, to prevent errors in high-impact scenarios.

Future work could involve ways to formalize these results in a more general manner, as well as developing tools to automate the mechanistic interpretability work, so that it is feasible on models with a large number of layers.

### LIMITATIONS

This work is limited by computational constraints. As a result of these constraints, the authors were unable to run further experiments on transformers with more layers, as well on a wide-range of tasks. Analysis is also avoided on state-of-the-art models due to the lack of compute.

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

## A    APPENDIX

This appendix contains results from the mechanistic interpretability techniques, model configuration, and various other visualisations for our model trained on 0-1 knapsack.

```
Output after Embedding: tensor([ 0.1288,  0.0006, -0.1608,  0.3533,  0.5096,  0.0870, -0.4737, -0.1977,
         0.2310, -0.3868, -0.0374], device='cuda:0')
Output after Attention: tensor([-0.2560, -0.0406, -0.1455,  1.2284,  1.4471,  0.3971, -0.7723, -0.0363,
        -0.6292, -2.8247,  1.8907], device='cuda:0')
Output after MLP: tensor([ 2.5303,  0.0191, -3.4691, 13.7954,  9.7629,  2.1123, -14.3343,
        -7.1281,  3.0515, -12.4285,  7.3215], device='cuda:0')
```

Figure 7: Results from logit lens

| Head | Weight_1 | Price_1 | Weight_2 | Price_2 | Weight_3 | Price_3 | Weight_4 | Price_4 | Capacity |
|---|---|---|---|---|---|---|---|---|---|
| 0.0 | 1.0 | 1.0 | 1.0 | 1.0 | -0.0044 | -0.0058 | -0.0138 | -0.006 | -0.0263 |
| 1.0 | 1.0 | 1.0 | 1.0 | 1.0 | -0.0051 | -0.0044 | -0.0038 | -0.0009 | -0.0258 |
| 2.0 | 1.0 | 1.0 | 1.0 | 1.0 | -0.0064 | -0.0109 | -0.0159 | -0.002 | 0.0067 |
| 3.0 | 1.0 | 1.0 | 1.0 | 1.0 | -0.006 | -0.0106 | -0.0144 | 0.0004 | -0.0252 |

Figure 8: Results from probing

| Layer | Index | Original Loss | Patched Loss | Change in Loss |
|---|---|---|---|---|
| 0.0 | -1.0 | 0.0 | 23.8995 | 23.8995 |

Figure 9: Results from activation patching

```
cfg = HookedTransformerConfig(
    n_layers = 1,
    n_heads = 4,
    d_model = 128,
    d_head = 32,
    d_mlp = 512,
    act_fn = "relu",
    normalization_type=None,
    d_vocab=cap+1,
    d_vocab_out=cap,
    n_ctx=3*n+1,
    init_weights=True,
    device=device,
    seed = 999,
)

num_epochs = 100000
```

Figure 10: Model configuration

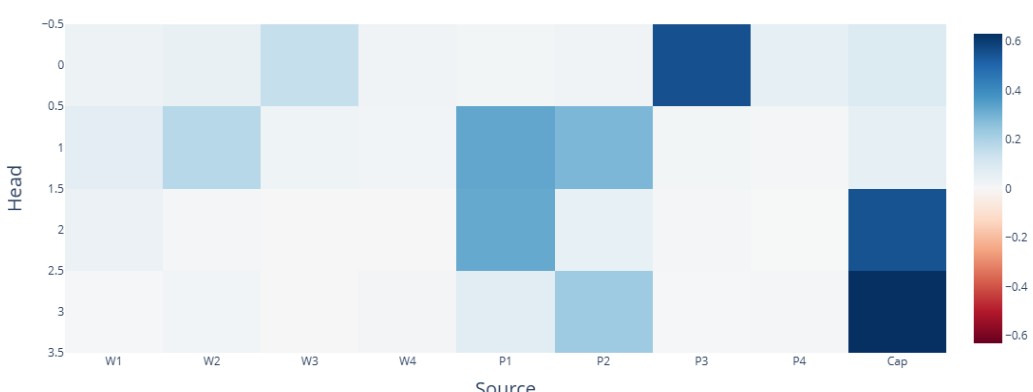

Figure 11: Attention pattern per head for a specific input sample

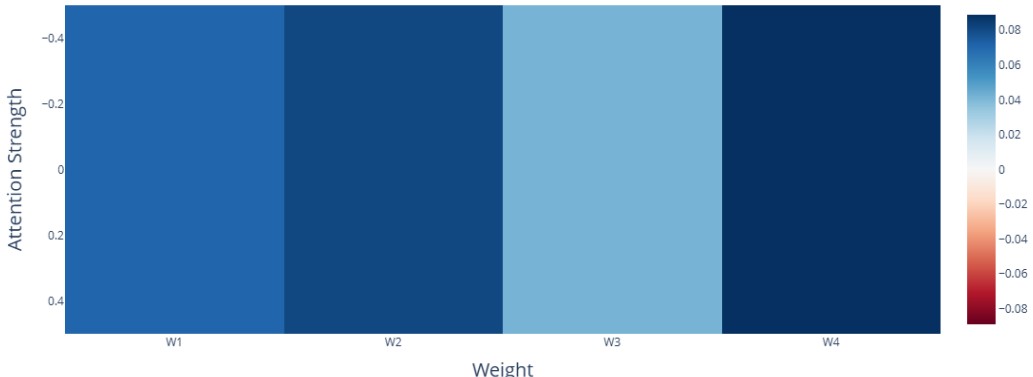

Figure 12:

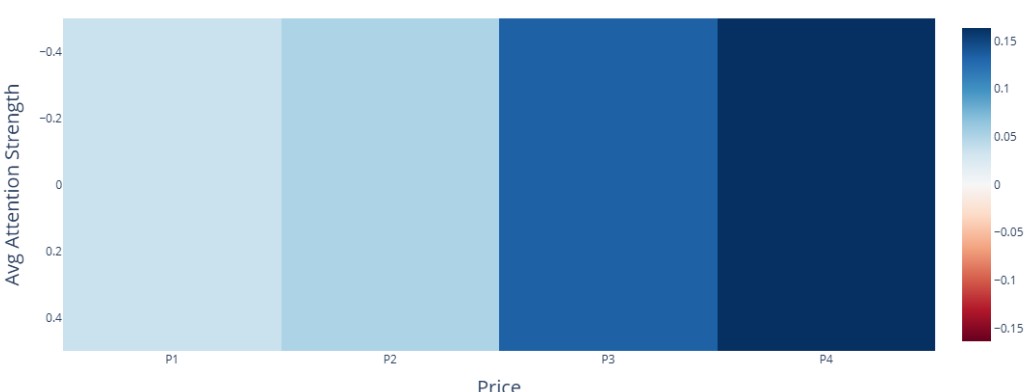

Figure 13:

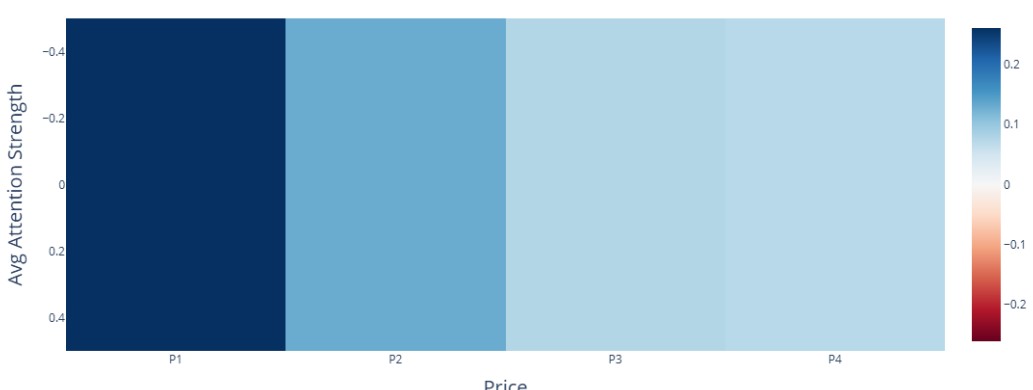

Figure 14:

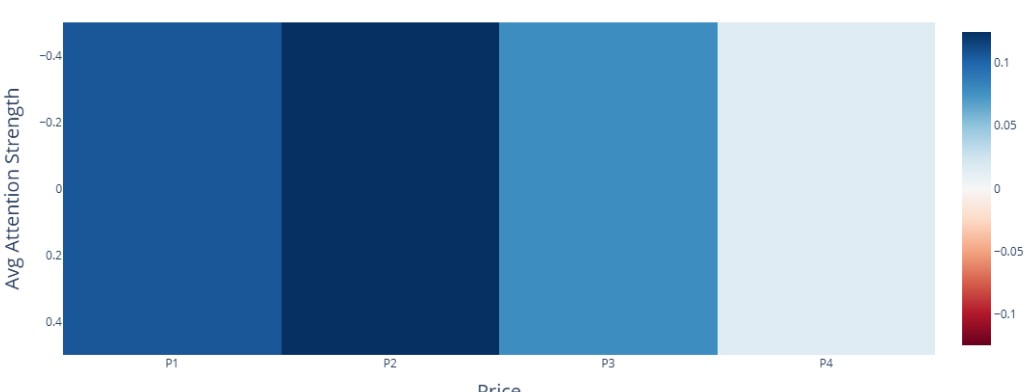

Figure 15:

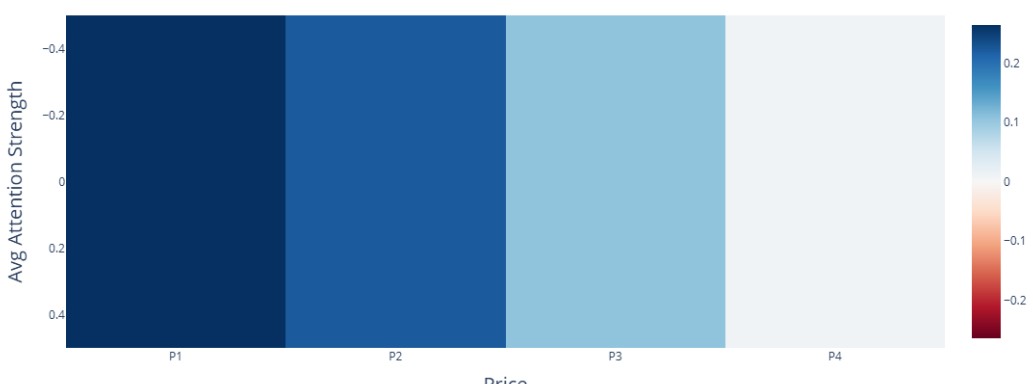

Figure 16:

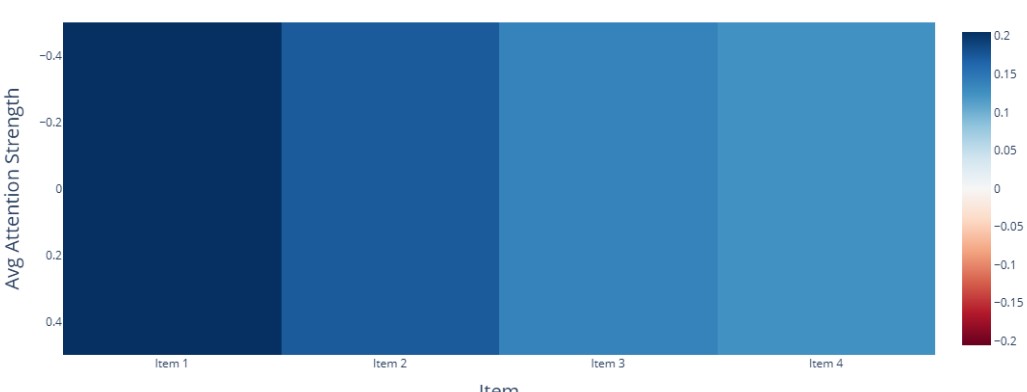

Figure 17:

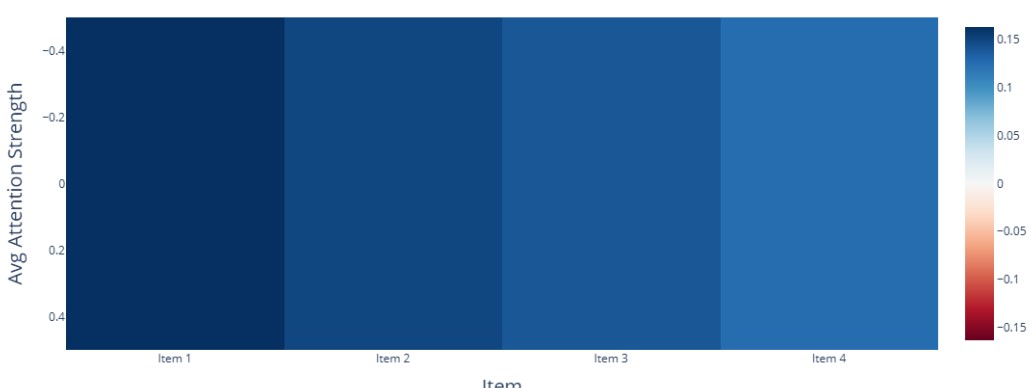

Figure 18:

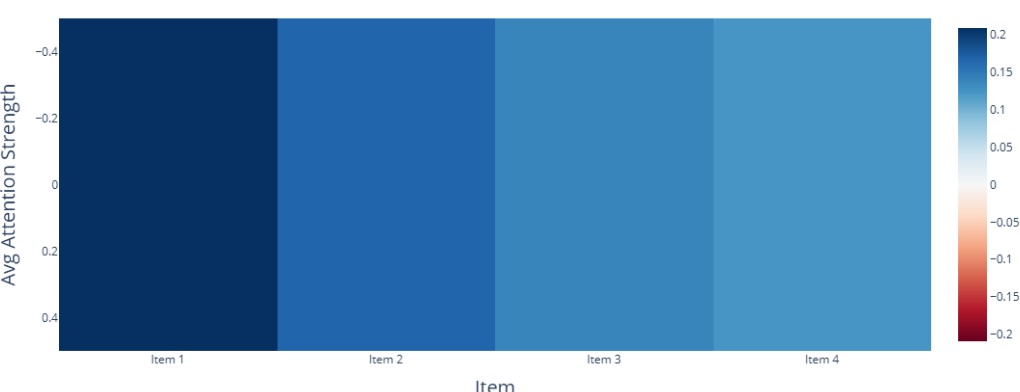

Figure 19:

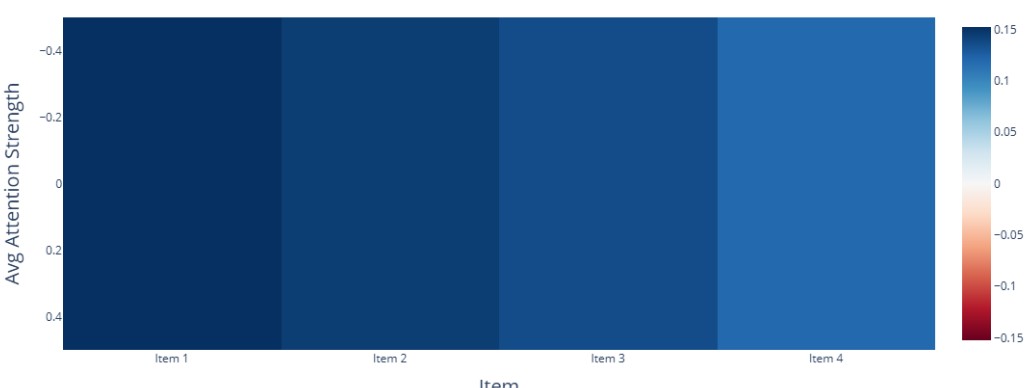

Figure 20:

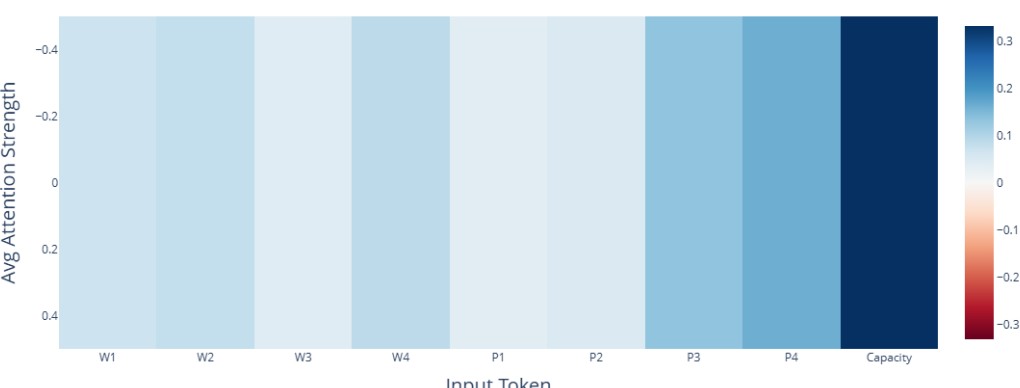

Figure 21:

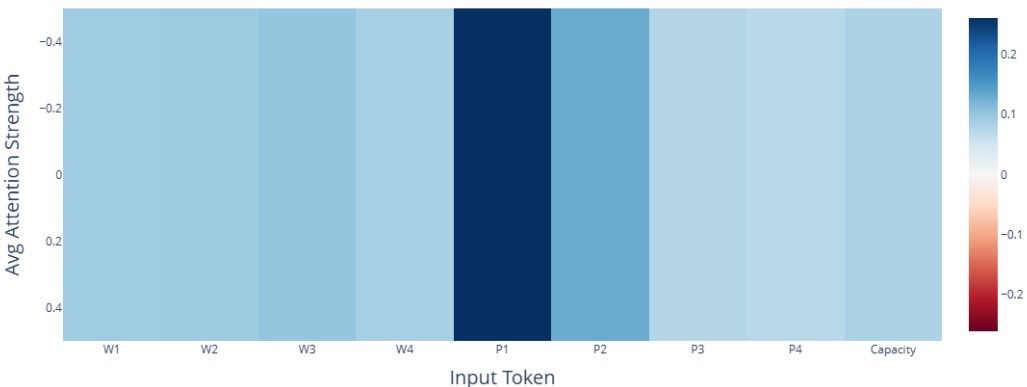

Figure 22:

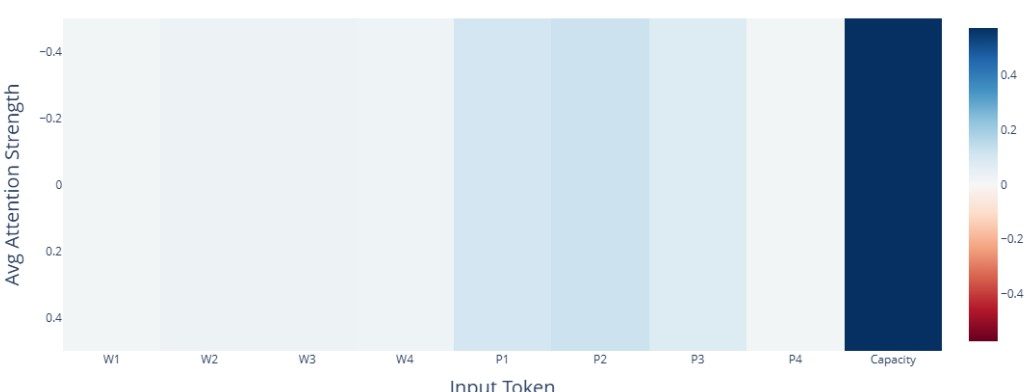

Figure 23:

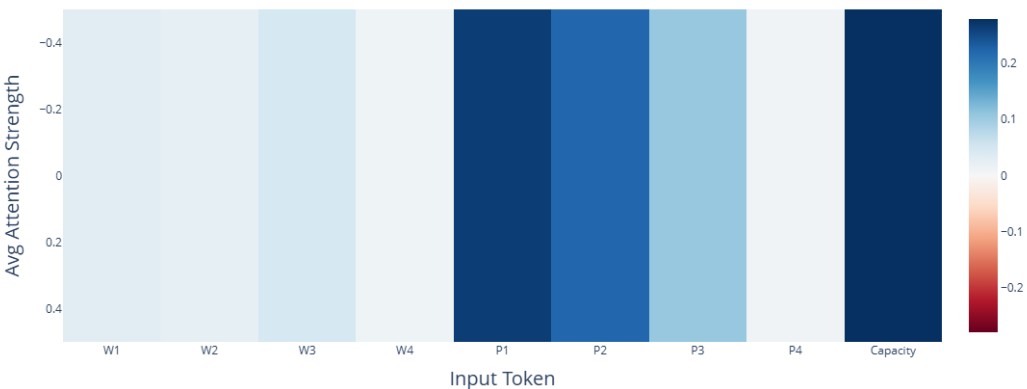

Figure 24:

## Principal Components on the Input

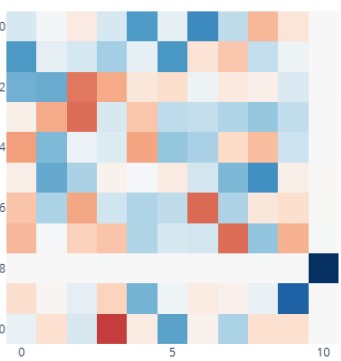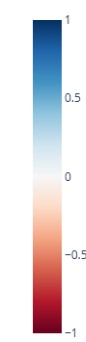

Figure 25:

## Principal Components Random

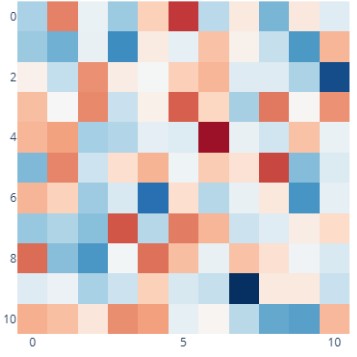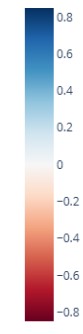

Figure 26:

## B  CODE FOR GENERATING DATASET

```python
for i in range(len(weights_permutations)):
    for x in range(n):
        temp[x] = weights[i][x]
    for j in range(len(prices_permutations)):
        for y in range(n):
            temp[n + y] = prices[j][y]
        for k in range(len(capacities)):
            temp[2*n] = capacities[k]
            best_picks, best_price = knapsack(weights[i], prices[j],
                capacities[k])
```

