# OpenReview forum: "Mechanistic Interpretability analysis of a single-layer transformer on 0-1 knapsack"
_ICLR.cc/2026/Conference — Submitted to ICLR 2026_

### Official Review · Reviewer_S3ND · 2025-10-31

**Soundness:** 1
**Presentation:** 1
**Contribution:** 1
**Rating:** 0
**Confidence:** 3

**Summary:**

This paper applies Mechanistic Interpretability (MI) to examine why a single-layer transformer fails to grok the 0–1 knapsack problem. The analysis reveals that the model relies heavily on the capacity token and the MLP layer, while its embedding space is poorly structured and resembles random initialization. Based on these observations, the authors argue that transformers have fundamental difficulty with NP-complete problems due to combinatorial explosion. They further hypothesize that a transformer with klayers can only generalize tasks with time complexity up to O(n^k), suggesting that large language models may face inherent limitations for high-stakes computational decision-making.

**Strengths:**

-The paper addresses an underexplored research area by examining how Transformers generalize on NP-complete problems.

**Weaknesses:**

-The paper arrives at the broad conclusion that transformer-based models struggle to generalize on NP-complete problems, yet supports this claim using only a single case study (the 0–1 knapsack problem) and only with a single-layer transformer. This limits the strength of the conclusion.
-The analysis is restricted to a synthetic dataset and a shallow model. To convincingly argue general limitations of Transformers on NP-complete tasks, the study should include additional NP-complete problems and architectural variants (e.g., deeper or pre-trained models).
-While the paper includes many figures, the accompanying explanations are insufficient, making it difficult to interpret the key findings from the visualizations.
-The work would benefit from a more formal theoretical examination of Transformer generalization in NP-complete settings, similar in spirit to prior analytical studies of model expressivity and computational limits [1, 2].
-The authors state that state-of-the-art models were excluded due to computational constraints. However, validating a claim with such broad implications requires experimentation on more capable models; otherwise, the conclusions may not generalize beyond the minimal architecture tested.

[1] Zaheer, M., Guruganesh, G., Dubey, K. A., Ainslie, J., Alberti, C., Ontanon, S., ... & Ahmed, A. (2020). Big bird: Transformers for longer sequences. Advances in neural information processing systems, 33, 17283-17297.
[2] Peng, B., Narayanan, S., & Papadimitriou, C. (2024). On Limitations of the Transformer Architecture. First Conference on Language Modeling.

**Questions:**

-What is the rationale for choosing the 0–1 knapsack problem as the case study? There are numerous NP-complete problems (e.g., n-SAT, Traveling Salesman, Hamiltonian Cycle) that may exhibit different structural characteristics. Clarifying this choice would help contextualize the scope of the conclusions.
-At which training stage(s) were the attention patterns visualized, and how do these patterns evolve over the course of training? Showing the progression could provide valuable insight into the model’s learning dynamics.
-The paper states that probing analysis shows the model “perfectly stores up to half of the weights and prices,” but detailed results or visualizations of this probing are not provided. Could the authors include these results to substantiate the claim?
-The hypothesis that a k-layer Transformer can only generalize tasks with time complexity up to O(n^k) is central to the paper, but the theoretical and empirical basis for this claim is not fully articulated. To make the hypothesis more testable and convincing, one possible experiment would be to vary model depth and examine generalization performance on a known polynomial-time task such as sorting—where a one-layer model would be expected to fail, while a two-layer model should succeed. This would provide direct support (or counter-evidence) linking model depth to computational complexity.

---

### Official Review · Reviewer_8m5D · 2025-11-01

**Soundness:** 1
**Presentation:** 1
**Contribution:** 1
**Rating:** 0
**Confidence:** 5

**Summary:**

This paper investigates why a small, single-layer transformer fails to solve NPC problem. The authors first construct a dataset of four-item 0-1 knapsack problem. Using the TransformerLens library, the authors trained a one-layer transformer. Through mechanistic interpretability methods—such as attention visualization, singular value analysis, logit lens, probing, and activation patching—they discovered that the model overemphasizes the knapsack’s capacity token, encodes information weakly, and lacks structured internal representations. These findings suggest that transformers struggle with NPC problems.

**Strengths:**

The research topic is important. Most of the difficult reasoning questions in the real world are NP-problems, like the Puzzle of 24.

The authors applied various mechanistic interpretability methods to interpret the model.

**Weaknesses:**

1. The paper may need to improve its presentation. For example, (1) explain the figures, (2) visualize the method and data, (3) make subsections, (4) split methodology and experiment.

Besides, to improve the flow, a general idea is not to use "we use xxxxx and we find xxxx", or "xxx is xxx, we use it and find xxx". A more easy-to-follow flow is first introducing the motivation, or the overview sentence of the paragraph or the subsection.

2. The author may need to add the discussion of related works and the discussion of the comparison with existing works. This paper is very relevant to LLM reasoning, LLM search/exploration.

3. The author may want to reconsider the scope of this work. I acknowledge that computation is a common issue, but considering more diverse tasks of NPC problems, or considering small-scale LLMs (e.g., 300M, 2B), is reasonable. The claim in the paper is strong, but the evidence is weak.

**Questions:**

See weaknesses.

---

### Official Review · Reviewer_GMDN · 2025-11-03

**Soundness:** 2
**Presentation:** 3
**Contribution:** 2
**Rating:** 2
**Confidence:** 4

**Summary:**

The paper studies whether a single-layer Transformer (via TransformerLens) can “grok” 0-1 knapsack when trained on an algorithmically generated dataset with 4 items per instance. The model is trained up to 100k epochs to predict the optimal total price given tokenized weights, prices, and capacity. It does not grok: train/test curves do not exhibit delayed generalization. The authors then run a suite of mechanistic interpretability probes, attention visualizations (capacity token receives most mass), logit lens (MLP dominates), linear probes (some weights/prices are recoverable; capacity poorly encoded), activation patching (capacity-focused neurons drive loss), and SVD/PCA on embeddings (spectra resemble random; unlike modular subtraction). They conclude that shallow Transformers struggle to form robust circuits for NP-complete tasks and hypothesize a depth-time-complexity link, i.e., k layers ≈ O(n^k ) algorithms.

**Strengths:**

1. Clean MI toolkit application (attention maps, logit lens, probes, activation patching, spectra).
3. Transparent reporting that the model fails to grok and that capacity dominates attention.
3. Honest limitations section and compute-aware motivation.

**Weaknesses:**

1. Fixed 4-item knapsack; no scaling across #items, model width/depth, or data size.
2. No MLP/RNN/DeepSets or symbolic/DP baseline; no depth>1 ablations.
3. Unclear train/test split; no out-of-distribution tests (e.g., new capacities, price/weight ranges).
4. Predicting only the optimal value sidesteps combinatorial structure (subset selection); may bias learning signals.
5. NP-complete generalization and O(n^k) depth hypothesis are unsupported by the presented evidence.
6. SVD/PCA on embeddings is suggestive but not decisive; no quantitative eval of probe accuracy vs layer/position.

**Questions:**

1. Why predict only the optimal value instead of the optimal subset (or value + subset)? Does predicting the subset change the behavior of grokking?
2. Report curves across items (4→6→8), depth (1→2→4), width, and data size. Where (if anywhere) does grokking emerge?
3. Add DP (optimal), greedy heuristics, MLP/DeepSets, and a 2–4-layer Transformer to separate depth vs optimization effects.
4. Specify train/test generation, show IID vs OOD generalization, and formal grokking metrics (e.g., Power et al.).
5. Provide causal tracing for capacity vs price/weight pathways and quantify probe R^2 /accuracy by token & layer.
6. Either theorize the O(n^k) connection or tone it down; a single 4-item, 1-layer experiment cannot support it.
7. Try factored inputs (set encoders / permutation-invariant layers) to test if failures stem from sequence order, not task hardness.
8. Compare classification over all feasible values vs regression vs program-of-thought target; report which helps.

---

### Official Review · Reviewer_B1LN · 2025-11-04

**Soundness:** 1
**Presentation:** 1
**Contribution:** 1
**Rating:** 0
**Confidence:** 5

**Summary:**

This paper applies mechanistic interpretability to a single-layer transformer trained on a small 0-1 knapsack dataset, finding it fails to generalize. It analyzes internals via visualizations and techniques, hypothesizing transformers struggle with NP-complete problems and are limited by layer count in computational complexity. Contributions aim to highlight LLM limitations in planning tasks.

**Strengths:**

The paper explores mechanistic interpretability on an NP-complete toy problem, extending beyond simpler prior tasks. It uses established tools to probe model failures, raising relevant safety concerns for AI agents in complex domains.

**Weaknesses:**

The claims about broad transformer limitations may be somewhat overstated given the constrained single-layer setup and small scale; exploring multi-layer models or larger instances could strengthen them. The methodology would benefit from additional baselines, clearer grokking criteria, and more ablation experiments. Presentation could be improved by addressing typos, providing more details, and better integrating figures. To improve, the experiments would have to be much more extensive, and any claims would have to have stronger backing from empirical evidence.

The mechanistic interpretability techniques are from previous literature, and don't offer any compelling evidence for the claims in the paper. It isn't strictly necessary for the paper to introduce new mechanistic interpretability techniques, but I'd recommend looking at [2] for an example of a stronger algorithm-oriented approach to interpretability.

It may be helpful for the authors to review work on tool-use. For instance, an argument can be made that even if LLMs cannot learn algorithms robustly, they can call tools to solve sub-problems, e.g. a knapsack solver. I personally think this argument has some flaws, but I'd encourage the authors to look into tool-use to strengthen their paper and claims.

The paper ignores work on transformer expressivity and algorithm learning, e.g. [1], which would make it clear that NP-complete algorithms shouldn't be able to be represented in a tiny transformer. Claims that transformers can learn simpler algorithms (abstract sentence 1, line 11), aren't really true [3], so it isn't necessary to jump to NP-complete problems. For example, transformers struggle to learn the parity algorithm effectively [2].

[1] Merrill, W., Sabharwal, A., & Smith, N. A. (2022). Saturated transformers are constant-depth threshold circuits. Transactions of the Association for Computational Linguistics, 10, 843-856.

[2] Shaw, P., Cohan, J., Eisenstein, J., Lee, K., Berant, J., & Toutanova, K. (2024). Alta: Compiler-based analysis of transformers. arXiv preprint arXiv:2410.18077.

[3] Markeeva, L., McLeish, S., Ibarz, B., Bounsi, W., Kozlova, O., Vitvitskyi, A., ... & Veličković, P. (2024). The clrs-text algorithmic reasoning language benchmark. arXiv preprint arXiv:2406.04229.

**Questions:**

I will use this section for a comment: While I do not recommend publication of this paper, I do not want to discourage the authors. I believe that their original intended idea is important to study, and hope to provide them with constructive options to explore for a future paper. My main comment is that it may be more valuable to study simpler algorithms, and connect the failures in representing these algorithms to common AI-agent approaches, e.g. to back up their claims in the abstract (lines 18-20).

---

### Meta-Review · Area_Chair_kfBc · 2025-12-14

**Summary:**

Learning how LLMs internally solve tasks is an important and timely research direction. However, this paper requires at least a major revision in the following aspects:

i) Related work review.
The paper states that Transformers cannot solve NP-complete problems, but this claim overlooks a substantial body of existing work on the expressiveness and computational limitations of Transformers. There are numerous papers studying related topics, from theoretical expressiveness results to formal limitations, which should be properly reviewed and contrasted with the claims made in this paper.

ii) Insufficient empirical support for the main claim.
Since the FLOPs of a Transformer scale polynomially, the community generally believes that Transformers cannot in general solve NP-complete problems. I appreciate the authors’ attempt to approach this question through interpretability analysis, but the current experimental evidence (e.g., the attention maps and linear probing results) is only on a simple 1-layer transformer and one NP-complete problem. This is not sufficient to substantiate the claim. Stronger and more systematic evidence is needed.

iii) Experimental setup needs substantial clarification.
The experimental setting should be explained in more detail in the main text, especially given that there is still ample space. Important aspects such as dataset generation, dataset size, model hyperparameters, and training details are missing. The current training and testing loss curves also appear indicative of overfitting.

iv) Figures and tables require significant improvement.
Figures and tables should be presented more clearly and accompanied by sufficient explanation. Some items in the appendix are direct bash outputs, which should not appear in a polished research paper.

v) Writing quality.
The overall writing could be improved for clarity, precision, and readability.

**Reviewer Concerns:**

There is no rebuttal, and I believe no concerns are addressed.

**Reviewer Scores:**

There is no rebuttal, all the reviewers will keep their origin scores.

---

### Decision · Program_Chairs · 2026-01-26

Reject